# Silicon as a Smart Fertilizer for Sustainability and Crop Improvement

**DOI:** 10.3390/biom12081027

**Published:** 2022-07-25

**Authors:** Rupesh Tayade, Amit Ghimire, Waleed Khan, Liny Lay, John Quarshie Attipoe, Yoonha Kim

**Affiliations:** Department of Applied Biosciences, Kyungpook National University, Daegu 41566, Korea or rupesh.tayade@knu.ac.kr (R.T.); ghimireamit2009@gmail.com (A.G.); waleedkhan.my@gmail.com (W.K.); layliny22@gmail.com (L.L.); jerryjohn2487@gmail.com (J.Q.A.)

**Keywords:** silicon, fertilizers, agriculture, sustainability, abiotic and biotic stress, Si transporters

## Abstract

Silicon (Si), despite being abundant in nature, is still not considered a necessary element for plants. Si supplementation in plants has been extensively studied over the last two decades, and the role of Si in alleviating biotic and abiotic stress has been well documented. Owing to the noncorrosive nature and sustainability of elemental Si, Si fertilization in agricultural practices has gained more attention. In this review, we provide an overview of different smart fertilizer types, application of Si fertilizers in agriculture, availability of Si fertilizers, and experiments conducted in greenhouses, growth chambers, and open fields. We also discuss the prospects of promoting Si as a smart fertilizer among farmers and the research community for sustainable agriculture and yield improvement. Literature review and empirical studies have suggested that the application of Si-based fertilizers is expected to increase in the future. With the potential of nanotechnology, new nanoSi (NSi) fertilizer applications may further increase the use and efficiency of Si fertilizers. However, the general awareness and scientific investigation of NSi need to be thoughtfully considered. Thus, we believe this review can provide insight for further research into Si fertilizers as well as promote Si as a smart fertilizer for sustainability and crop improvement.

## 1. Introduction

Silicon (Si) is a metalloid element and is the most commonly found element in the earth’s crust after oxygen [1]. Orthosilicic acid is the plant available form of silicon present in a small amount in the soil, which is absorbed by plant roots at pH < 9, and its concentration ranges between 0.1 to 0.6 nM [2]. The effect of Si application on several plant species has been proven for several decades, and research has shown numerous advantageous effects of Si on different plant attributes including vegetative growth, development, production, and reduction of biotic and abiotic stress [3,4,5,6]. Other Si-driven benefits include increase in photosynthetic rate, lower transpiration rate, and resistance to UV-B radiation [7].

Plants face several biotic and abiotic challenges, such as pathogenic infections, insect pest damage, drought, flooding, salinity, and metal toxicity. The availability of Si in plants is inadequate in most agricultural soils [2,8,9]. Hence, Si is commonly supplied as a fertilizer to assess the impact on plants. Vegetable and fruit crops, some dicotyledonous plants, and monocotyledonous plants can actively absorb and accumulate Si at high levels in their plant tissues [1,10]. Application of Si fertilizer takes into account soil fertility, crop needs, soil moisture patterns, and crop uptake of nutrients from the soil, also help in maintaining the ratio of nutrients withdrawn by crops in check [4]. The utilization of Si fertilizers has resulted in increased plant growth rate, higher productivity, protection against a wide range of pathogen invasions, and adaptation to unfavorable environmental conditions [2,5,6]. For example, in rice (*Oryza sativa*), application of Si fertilization resulted in increases in the culm wall thickness, vascular bundle size, and peroxidase activity, thereby increasing stress tolerance, stem strength, and preventing lodging [11,12]. Similarly, Si supplementation as a fertilizer tends to alter level of proline in upland rice cultivars during water stress in the vegetative and reproductive stages; this effect could be indicative of stress tolerance [13]. In addition, supplementation with Si is thought to increase the water flow rate in the xylem vessels and improve water use efficiency by reducing water loss through transpiration in plants under drought stress in maize (*Zea mays*) [14,15,16]. Murillo-Amador et al. (2007) [17] reported that sodium chloride (NaCl) stress in cowpeas (*Vigna unguiculata*) and kidney beans (*Phaseolus vulgaris* L.) caused significant reductions in plant growth; however, Si supplementation greatly improved their growth by increasing net photosynthesis, chlorophyll content, stomatal conductance, transpiration, and intercellular carbon dioxide [14]. Si supplementation improved the oxidative defense system of waterlogged crops, thereby enhancing their tolerance to flooding). Moreover, Si induces a defense against different insects and pests, including sugarcane stalk borers, leaf spiders, stem borers, green leafhoppers, brown planthoppers, and mites [18,19,20]. Conclusively, Si-based compounds have multiple and diverse uses in crop production in the agricultural sector.

Globally, crop species face challenges from climate change, and abiotic and biotic stresses lead to significant crop yield reductions. The current crop production trend is not sufficient to meet the growing demand driven by the population rise by 2050 [21]. Therefore, numerous management technologies are required to boost crop productivity. It should be noted that sustainability is vital for the future of agriculture as well as the planet. In this context, the improvement of fertilizers, such as Si fertilizer application, can greatly influence plant attributes and aid in improving plant health and yield. The impact of Si fertilizer on crop growth, yield, and quality is well recognized. Si fertilizers (organic and inorganic) are high-quality fertilizers considered eco-friendly and sustainable for agriculture because of their noncorrosive and pollution-free nature; they can also be inexpensive and thus be considered “smart fertilizers.” In addition, silicate fertilizers are used for soil pH correction and in macro- and micronutrient acquisition, particularly when slag or Si-containing mineral ores are utilized as silicate fertilizer sources [22].

In this review, we present an overview of the different smart fertilizers; application of Si fertilizer in agriculture (monocotyledonous and dicotyledonous plants); availability of Si fertilizers; experiments conducted in greenhouses, growth chambers, and open fields; and prospects to promote Si as a smart fertilizer among farmers and the research community for sustainable agriculture and yield improvement.

## 2. Smart Fertilizer Types

In general, smart fertilizers are fertilizers applied in soil that allow for managing the rate, timing, nutrient release duration, and active absorption by the plant roots [23]. Recently, Raimodi et al. (2021) [24] defined a smart fertilizer as “any single or composed nanomaterial, multicomponent, and/or bioformulation containing one or more nutrients that can adapt the timing of nutrient release to the plant nutrient demand via physical, chemical, and/or biological processes, thereby improving crop growth and development and reducing environmental impact when compared to conventional fertilizers.” Smart fertilizers are classified as (i) nano fertilizers, (ii) composite fertilizers, and (iii) bioformulations [24].

Nano fertilizers, which are based on nanoparticles, as their name suggests, are available as powder or liquid formulations, including manufacturing, design, and application [24]. These fertilizers can help improve nutrient release kinetics and plant uptake efficiency [25], resulting in benefits including increased crop productivity [26,27,28,29], decreased nutrient loss to the environment [30] and enhanced nutritional quality and shelf life [31].

Composite fertilizers are composed of various materials containing one or more nutrients that have been developed to take advantage of material synergy and improve essential plant nutrition [24,32]. Based on material properties, organic and inorganic coating materials (including granules and hydrophobic or hydrophilic materials [matrix or gel]) and inorganic compounds with low solubility are commonly used for coating or mixing fertilizers [24,33,34,35,36,37].

Bioformulations are fertilizers containing active or dormant microorganisms, such as bacteria and fungi capable of influencing physiological growth responses, nutrition, and protection improvement in plants [38].

The precision of fertilizer usage has become a vital point in the agricultural sector worldwide because of the need for increased productivity and reduced over-fertilization with negative impacts on the agricultural ecosystem [39]. From these viewpoints, smart fertilizers have been utilized to boost agronomic fertilizer efficiency while lowering environmental impact and fertilizer costs. Consequently, several novel fertilizers have been developed.

## 3. Effects of Si Fertilizers on Crops under Controlled Environmental Conditions

Si uptake and translocation in plants are influenced by the type of plant, the type of root system, and the chemical composition of the soil [40]. In the soil-plant system, Si fertilizer has a bifurcated impact. First, it strengthens the plant’s protective properties against biotic and abiotic stress. Second, the addition of biogeochemically active Si compounds to soil improves the soil’s physical, chemical, and water content, as well as the availability of nutrients in plant-available forms [41]. Si fertilizer application has a positive impact on plant growth, mineral nutrient balance, and lodging. Foliar application of potassium silicate and magnesium silicate fertilizers improved the nutrient content and yield in flax [42]. Si supplemented plants were also observed to have improved root characteristics, such as total root length, root surface area, and lateral root length, thus inducing plant resistance to drought [43] and element deficiency conditions [44]. Despite the large differences in Si uptake among rice, tomato, cotton, onion, and chili, amorphous Si fertilizer showed improved productivity in all crops [45].

Globally, 50% of crop loss occurrence is due to abiotic stress [4,46]. Si fertilizers have a substantial impact on abiotic and biotic stress tolerance, including drought stress, temperature tolerance, UV tolerance and disease/pest resistance respectively. Reportedly, Si fertilization in grapevines elicited an induced chemical response to pest attack and induced defenses against pests [47]. According to Lemes et al. (2011), the utilization of Si fertilizer decreased the quantity and rate of fungicide application, lowered production cost, and optimized control practices in areas where Si was naturally low.

Studies on monocotyledonous and various dicotyledonous plant species in greenhouses/growth chambers have demonstrated that Si fertilizer increases resistance level against biotic stress, including lepidoptera, hemiptera, homoptera, diptera, thysanoptera, oleoptera, and non-insect pest attacks [19,48,49,50]. Si fertilizer also reduces the susceptibility of many monocotyledonous and dicotyledonous plant species to fungal diseases [51]. However, there have been some contradictory reports suggesting no significant differences in the control of gray leaf spots. For example, Nanayakkara et al. [52] conducted experiments in a controlled environment using two sources of Si supplement: calcium silicate and wollastonite (calcium inosilicate). The authors found that Si sources had no significant differences in the control of gray leaf spot of perennial ryegrass turf compared with control rice plants that absorbed an ample amount of Si (up to 10% of shoot dry weight) in the form of monosilicic acid, which further accumulated in the form of a cuticle-Si double layer [53]. Monocotyledonous and dicotyledonous plants have a large number of Si transporters (influx and efflux). The Si uptake mechanism differs across monocotyledonous and dicotyledonous plants because of the differences in their root architecture [54]. Monocots can accumulate Si up to 10% of the plant mass, whereas most dicots can accumulate less Si [42,55,56,57]. The following families increase Si uptake and utilization: among the monocotyledonous plants, *Poaceae* (*Gramineae*) and *Cyperaceae* [58], and among the dicotyledonous plants, *Fabaceae*, *Cucurbitaceae*, *Rosales*, and *Asteraceae*. However, the shoots of dicotyledonous plants are unable to gather high levels of Si. In a perennial dicotyledonous strawberry plant study, the greenhouse experiment showed an increase in vegetative growth and fruit dry weight after Si fertilization [59]. 

Studies on numerous monocotyledonous and dicotyledonous crop plants have shown that Si has beneficial effects; these effects of Si are more obvious when the plants are under stress (Table 1). Overall, the effect of Si-based fertilizers on different plant species is well established, but the mechanisms underlying their beneficial effects are still elusive. In Figure 1, we illustrate the mechanism of Si uptake and transport from the roots to the different plant parts. The ways by which Si uptake and transport influence the other general mechanisms of plants, which result in altered plant attributes, including abiotic, biotic stress tolerance and yield enhancement, are also presented in this figure.

## 4. Application of Si under Field Condition

Generally, intense weathering, leaching, and intense agricultural practices promote desilication and Si removal from soil. Fertilizers are critical to the global food supply. Farmers typically utilize individual commercial chemical fertilizers to meet the nutrient needs of plants. As they are cost-effective and provide quick results, such fertilizers are widely used by farmers [94]. A positive effect of Si fertilization on some essential plant nutrients has also been reported. Silicon fertilization has a dual effect on the soil-plant system. First, it improves plant Si nutrition and increases plant resistance against diseases, insect attacks, and unfavorable environmental conditions, such as drought, salt, heavy metals, and hydrocarbon toxicity. Second, soil treatment with biogeochemically active Si substances optimizes soil fertility through improved water, physical, and chemical soil properties and maintenance of nutrients in plant-available forms. On average, Si fertilization increases the yield by 10–25% [41]. When Si was applied to barley grown under flooding conditions, it reduced the intensity of oxidative damage without any significant change in the antioxidant activity; a reduction of nitrous oxide production in the flooded soil was also observed [95]. Likewise, another source of Si—potassium silicate (K_2_SiO_3_)—when sprayed on the leaves of maize plants, increased water use efficiency and yield under low irrigation conditions as well as enhanced protein and oil contents of the grain [96]. In one study, the application of Si fertilizer increased N, P, K, Ca, Fe, Mn, Cu, and Zn concentrations. Sugarcane parameters showed a positive relationship with Si in plants; the height and stalk diameter and the dry leaf biomass were 50, 58, and 71% higher compared with those of the control plants. In addition, damage caused by the stalk borer was significantly reduced because of the accumulation of Si [97]. Different sources of Si fertilizers are available for application in the field, as shown in Figure 2.

## 5. Various Si Sources for Si Fertilizer Application

Industrial wastes such as blast furnace slag, steel slag, fly ash, and bottom ash are known to be potential silica fertilizers [100,101]. Particularly, steel slag is a by-product material formed during the process of manufacturing steel from iron, whose volume accounts for 15–20% of the total steel produced [102]. The basic properties of slag vary according to the manufacturing, cooling, and valorization processes [103]. The nutrient constituents of steel slag are calcium oxide CaO (40–50%), silicon dioxide SiO_2_ (10–28%), iron Fe (17–28%), traces of magnesium oxide MgO (2.5–10%), and manganese oxide MnO (1.5–6%) [104]. These slags, produced as byproducts, have been used in agricultural fields as Si supplements. However, only a few studies have been conducted to distinguish the effects of different slag-based fertilizers on plant growth and disease incidence. For example, steel slag- and iron slag-based Si fertilizers were used to determine their effects on yield parameters and disease incidence in rice [105]. The authors reported that steel slag showed a stronger effect than iron slag in increasing the yield and lowering the incidence of brown-spot disease in rice.

Approximately 80 million tons of rice husk/hulls is produced worldwide, which comprises nearly 3.2 million tons of Si, mainly in the form of ‘opaline silica’ [106]. When converted into biochar under partial oxygen, this nutrient-rich rice husk is used as an effective soil ameliorant [107]. Rice husk biochar (RHB) decreased the level of insect infestation, while simultaneously increasing the level of Si in leaves of brinjal plants [108]. Similarly, when used together, Si-rich rice husk ash (RHA) and RHB lowered the amount of inorganic arsenic buildup in rice grains [109]. The studies also concluded that Si obtained from sources such as RHB and RHA would be available in a more soluble form than Si from silicate minerals. Previously, it was reported that K_2_SiO_3_ also developed resistance to disease incidence, and its application suppressed powdery mildew in strawberries [110] and mildew in tomato plants [111]. In addition, foliar application of another source of Si, sodium silicate (Na_2_SiO_3_), on maize increased the chlorophyll and carotenoid contents and the uptake of potassium and calcium by plants under drought stress conditions [112]. Another study evaluating the effects of Na_2_SiO_3_ in response to disease occurrence showed that the use of Na_2_SiO_3_ lowered Fusarium wilt in cucumbers by changing the soil microbial properties and increasing the seedling growth of the plant [113]. Application of monosilicic acid when the concentration of heavy metals was high in barley plantation soil showed that with an increase in the acid concentration from 5–20 mg kg^−1^ to ≥50 mg kg^−1^, there was a reduction in the mobility of heavy metals in plants and an increase in plant resistance against these heavy metals [114]. Likewise, when monosilicic acid was used in rice against cadmium toxicity, toxicity levels were reduced in roots, leaves, and stems by 50–90% through the apoplast pathway of cadmium transportation [115].

### 5.1. Application of Different Commercially Available Silicon Fertilizers

Hydrophilic fumed silica (SiO_2_), produced by Evonik Industries under the commercial name “Aerosil 200” (average particle size = 12 nm), improved the seed germination, increased chlorophyll content and photosynthetic rate, and improved the growth parameters of squash (*Cucurbita pepo*) under salt stress condition [116]. Diatomaceous earth, produced by Agripower Australia Private Limited, when applied at a rate of 600 kg per hectare along with standard fertilizer practice, resulted in a significant increase in the yield and nutrient uptake in rice [117]. Soluble Si, in the commercial name Agribooster^TM^, significantly increased the chlorophyll content in onion plants, proportionately increasing the photosynthesis rate under environmental stress conditions [118]. The same soluble Si fertilizer used in soybean increased water availability, thereby increasing the peroxidase enzyme, which has scavenging roles against stress conditions [119]. Similarly, a commercial silicon fertilizer produced by Pungnong Co., Ltd. (Seoul, South Korea) consisting of SiO_2_, MgO, and CaO (25%, 2%, and 40%, respectively), when applied to soil along with a foliar application of sodium metasilicate, enhanced the root morphological traits of soybean plants. Its application also increased the nodule number, nodule size, and photosynthetic parameters, which ultimately increased the yield by 21% and 19% in two (2018 and 2019) different years [120]. In the field or greenhouse numerous Si sources (solid and liquid) have been used as fertilizers for agronomic and horticultural crops which are produced and marketed by commercial enterprises (Table 2).

### 5.2. Application of NanoSi in the Field

Nanotechnology is a revolutionary field with considerable potential in the agricultural sector, particularly in the context of yield enhancement and plant disease management [121]. Nanotechnology is a new platform for increasing global food production, which has become a hot research topic for every sector involved in agri-food production, and its applications have opened a new research field in biotechnological and agricultural science [122]. NanoSi (NSi) has been reported to contribute to the growth and yield of plants under different biotic and abiotic stress conditions and in the effective utilization of the fertilizers applied [123].

A significant increase in harvested and milled dry grains was observed in a field experiment of black rice when 1 N of NSi was used along with N, P, and K [124]. A similar result was observed in two Iranian rice varieties, where the maximum harvest index was obtained using a combination of NSi and nitroxin [125]. Consistently, the application of NSi (with 2% chelated Si) increased the yield of wheat by 28% and 32% in two different years of field experiments under deficit irrigation conditions [126]. The same experiment showed that water use efficiency was maximum in both years under full irrigation and deficit irrigation conditions upon foliar application of NSi. In cucumber, an increase in the growth parameters and yield was observed in a field experiment using NSi fertilizer at a rate of 60 mg/L under salt stress conditions [127]. Several NSi fertilizers have been used to assess their effects on diverse plant species (Table 3).

In summary, the role of Si under stress conditions is prevalent indeed, either through direct or indirect mechanisms (not proven at the molecular level); thus, Si fertilizer can play an important role in improving production and maintaining sustainability in the agriculture sector. However, several questions remain regarding the effective use of Si fertilizers.

## 6. Conclusions and Future Perspectives

A comprehensive review of the literature revealed that Si and its fertilizer form can stimulate plant growth and alleviate various abiotic and biotic stresses in several crops. However, there is much scope for further research on developing new Si-based fertilizers using advanced nanotechnology approaches. Our literature survey revealed that the molecular mechanism of Si under control and stress conditions is critical for improving our understanding. Thus, we suggest the following points for this prospect.

Fertilizers are vital for global food production, and the use of Si-based fertilizers in agriculture is expected to increase in the future. This is because intense cultivation of crops over the years has depleted Si and the soil nutrient status of many agricultural soils. In addition, the increased use of chemical fertilizers and pesticides causes environmental and ecosystem degradation, biodiversity loss, and public health risks, which ultimately lead to huge economic losses and sustainability crises. Si accumulator plants account for seven of the top ten most important crops worldwide, implying that crops other than Si accumulators will almost certainly require Si applications in the future to maintain optimal yields. That Si is eco-friendly, prophylactic, and plays an important role to improve soil health and plant resistance to biotic and abiotic stresses through diverse mechanisms is now well established. Several reviews have been published on pure Si but not on Si fertilizers, and most of them suggest that the mechanism of Si in plants is still unclear. Recently, Coskun et al. (2019) [138] found that many of the molecular assumptions regarding Si activity contradict most of the scientific evidences, particularly those acquired from current functional genomics. Thus, future research should focus on the following:

(i) Understanding the precise molecular basis of pure Si in the biological activity of diverse plant species.

(ii) Demonstrating the beneficial role of pure Si as well as Si fertilizers under stress conditions. Very few studies have been conducted under normal conditions; thus, efforts should be invested to determine the effect of Si or Si fertilizer without inorganic ions, such as Mg and Ca, under normal conditions.

(iii) Ensuring beneficial outcomes. Determining the precise application dose/rate of Si fertilizers is crucial; hence, further research should be consistently conducted for crop-specific Si accumulators and pH of the soil.

(iv) Nanotechnology has the potential to greatly influence the future of agriculture. Despite this, very little work has been conducted to utilize nanotechnology in agriculture; thus, novel nanoparticle approaches should be explored further to develop new NSi, which can enhance the catalytic performance of Si fertilizers. Focus should also be given to investigating particle integrity, efficiency, movement, toxicity, and sustainability of newly developed materials.

(v) After utilizing the new NSi at preliminary studies (to determine feasibility, time of release, cost, scaling factors, unpredicted results), its consistency and performance should be evaluated in large or actual fields.

As a consequence of climate change, major and minor crops globally are likely to face new constraints and yield penalties as well as prospects for sustaining productivity. Si fertilizer application is a common agricultural practice in several regions; however, global awareness is important for promoting the potential use of Si fertilizers and minimizing the gap between researchers/developers and farmers. We consider that this review may provide insights into Si fertilizers and their applicability in agriculture as a long-term strategy for mitigating biotic and abiotic stresses in the future.

## Figures and Tables

**Figure 1 biomolecules-12-01027-f001:**
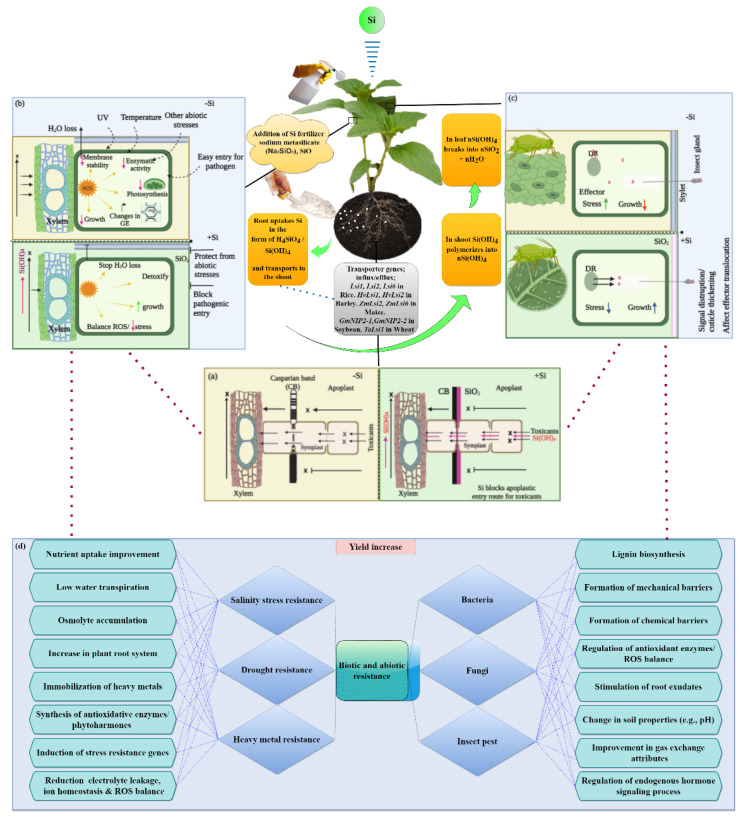
Mechanism of Si-induced apoplastic obstruction and its beneficial effect on the applied plants. (**a**) Conditions before (−Si) and after (+Si) Si application; Si blocks the apoplastic entry route of toxicants, which is the common pathway in addition to the symplastic route to reach the xylem. Casparian bands (CBs) are present, stopping the entry of toxicants, although breaks may occur allowing for bypassing routes, particularly under low or −Si conditions. High or +Si improves CB development [76], as well as apoplastic Si deposition (as silica, SiO_2_ [77], effectively blocks bypass routes resulting to root-to-shoot translocation of toxicants). (**b**) In +Si plants, when Si translocates to the plant shoot cells, it deposits in the form of SiO_2_, forming a thick layer in the apoplastic region or a mechanical barrier blocking the pathogenic entry, and protects from other abiotic stresses. Furthermore, it modulates reactive oxygen species (ROS) level leading to detoxification and reduction of stress effect and increased plant growth. However, in the case of −Si plants, toxicant levels in shoots accumulate to a greater extent causing adverse effect on plants. (**c**) In +Si plants, the formation of an SiO_2_ barrier layer in the apoplastic region disrupts the process of establishing specificity between a plant and insect by altering the flow of molecules (e.g., effectors). This also helps the plant to prevent piercing by insects, resulting in the limited translocation and release of effectors in +Si plants compared with that in −Si plants [78]. (**d**) The overall effect of Si application influences the other general mechanisms of plants, altering plant attributes, including abiotic and biotic stress resistance, which leads to yield enhancement. For further details about the general mechanisms, see the references in [79,80,81,82,83,84,85,86,87,88,89,90,91,92,93].

**Figure 2 biomolecules-12-01027-f002:**
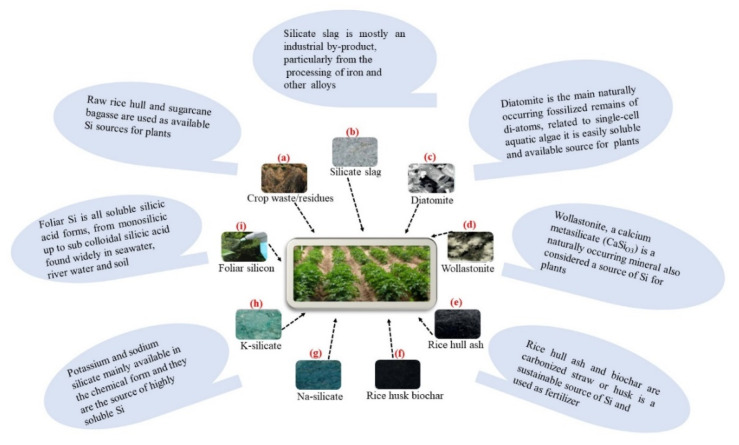
Different sources of Si fertilizer used in agricultural applications. (**a**) Crop raw rice hull, (**b**) silicate slag, (**c**) diatomite, (**d**) wollastonite, (**e**) rice hull ash, (**f**) biochar carbonized straw or husk, (**g**) sodium silicate, (**h**) potassium silicate, and (**i**) foliar Si soluble silicic acid forms, right from monosilicic to subcolloidal silicic acid [98,99].

**Table 1 biomolecules-12-01027-t001:** Effects of silicon fertilizers on monocotyledon and dicotyledon crops.

Crop	Growing Condition	Stress	Type of Si Fertilizer Used	Effect on Plant	References
Cucumber	Greenhouse	Salt stress	Potassium silicate (K_2_SiO_3_)	Decreases oxidative damage of plant tissue under salt stress conditions; enhances physiological and metabolic activities	[60]
Sugarcane	Shade house	Stem border	Calcium silicate (CaSiO_3_)	Less stalk damage by reducing the growth of stem borer larva and delaying its penetration to the stalk	[48]
Grasses (Timothy, Sudan, Rhodes, and Tall fescue grass)	Greenhouse	Water stress	K_2_SiO_3_, CaSiO_3_	Increase in biomass yield and enhance nutrient uptake	[61]
Canola	Greenhouse	Salinity stress	Sodium silicate (Na_2_SiO_3_)	Improves plant growth and saline tolerance	[62]
Soybean	Greenhouse and fields	Fungal disease	CaSiO_3_ and K_2_SiO_3_	Suppress disease development; delay disease impact during the growth cycle	[51]
Sunflower	Greenhouse	Caterpillar	Si + acibenzolar-S-methyl (ASM)	Increase the lignin and Si content in plants, thus inhibiting caterpillar development	[63]
Water spinach	Control chamber	Arsenic	Iron and silicon fertilizer	Enhance plant growth and reduce the arsenic taken up by the plant, thereby reducing human health risks from consumption	[64]
Potato	Greenhouse	Salinity stress	Hydrophilic bentonite (H_2_Al_2_O_6_Si)	Improves root characteristics to cope with drought and nutrient deficiency; increases tubers	[43]
Strawberry	Shade house	Water deficit	K_2_SiO_3_	Improves plant growth and development	[65]
Wheat	Greenhouse	Heavy metal toxicity	Organosilicon fertilizer (OSiF)	Promotes wheat growth and increase biomass; enhances photosynthetic parameters and chlorophyll content; reduces oxidative damage and content of cadmium in roots and lead in shoots, branches, and flowers	[66]
Tomato	Greenhouse	Drought-induced reactive oxygen species	Na_2_SiO_3_	Promotes energy consumption in mitochondria, thus increasing photosynthesis	[67]
Red-lettuce	Chamber, Hydroponic	Light stress	Liquid silicon-containing fertilizer	Increase in yield biomass for red-leaved lettuce under LED lighting	[68]
Grape tomato	Greenhouse (Polyhouse)	Drought	Monosilicic acid (H_4_SiO_4_)	Increases fruit number and yield; enhances irrigation water productivity	[69]
Cantaloupe (Rock melon)	Polyhouse	Drought	H_4_SiO_4_	Improves fruit quality and flesh thickness under moderate drought and well-watered conditions	[70]
Rice	Greenhouse	Drought stress	K_2_SiO_3_	Improves photosynthesis, photochemical efficiency, and mineral nutrient level	[71]
	Greenhouse	Sugarcane borer	Soil Si amendments	Decrease in feeding injury; attracts natural enemies	[72]
	Greenhouse	Leaf folder	Si-mediated	Resistance to leaf folders for the susceptible varieties	[73]
	Greenhouse	Cadmium and lead	Organosiliconeand mineral silicon fertilizer	Reduces rice uptake of cadmium and lead contamination; increases grain yield for brown rice; improves the antioxidant capacity of rice; alleviates stress from heavy metals	[74]
	Chamber	Herbivores	Na_2_SiO_3_	Enhances the attractiveness of herbivore-induced plant volatile blends	[75]

**Table 2 biomolecules-12-01027-t002:** List of commercial sources of Si products and relevant organizations.

Company/Institute	Product Name	Product Description	Website
Agripower	Agripower Silica/Agrisilica/Agrisilica Chip Silicon Fertilizer Silicon Based Natural Plant Nutrient	Diatomaceous earth	https://agripower.com.au/ (access on 26 May 2022)
Agriculture Envision US Inc.	Blue Silicate	Fertilizers and soil amendments, blended	https://www.omri.org/mfg/aeu/ (access on 26 May 2022)
CalSil Corp.	Sili-Cal	Calcium silicate	http://www.calsilcorp.com/ (access on 26 May 2022)
Canadian Wollastonite	Canadian Wollastonite	Calcium silicate soil amendment	www.canadianwollastonite.com/ (access on 26 May 2022)
Dyna-Gro	Pro-Tekt	Potassium silicate solution	https://dyna-gro.com/ (access on 26 May 2022)
Eden Solutions	Blue Gold Silica	SiO_2_	https://edenbluegold.com/product/blue-gold-silica-omri-listed/ (access on 26 May 2022)
Haitor Silicon Fertilizer Research Institute	Nanosilicon fertilizer, Seaweed Silicon, Water Soluble Silicon, Potassium Silicate Foliar Fertilizer, Silicon Calcium Magnesium Fertilizer, Soil conditioner	SiO_2_, CaO, MgO	https://www.haitoragro.com/silicon-fertilizer/silicon-solid-fertilizer/nano-silicon-fertilizer.html/ (access on 30 May 2022)
Harsco	CrossOver	Calcium and magnesium silicates	https://www.crossover-soil.com/ (access on 30 May 2022)
Montana Grow	Natural Si Fertilizer	Diatomaceous earth	https://montanagrow.com/ (access on 30 May 2022)
Plant Tuff Inc.	Plant Tuff	Calcium and magnesium silicates	http://planttuff.com/ (access on 30 May 2022)
Pilares Operations	IMERYS WOLLASTONITE Calcium SilicateNYAD^®^ MD200	Diatomaceous earth, wollastonite	https://www.imerys.com/minerals/ (access on 30 May 2022)
Privi Life Sciences Ltd.	Silixol	Orthosilicic acid	http://privilifesciences.com/ (access on 30 May 2022)
PQ Corp.	Sil-MATRIX	Potassium silicate	https://www.pqcorp.com/ (access on 30 May 2022)
Rexil Agro	OSAB3	Hydrated silicon dioxide, potassium chloride, polyethylene glycol, and boric acid	http://rexil-agro.com/ (access on 30 May 2022)
TEAA Tecnología Agrícola Avanzada SA de CV	SIK Silicato de Potasio	Potassium silicate, aqueous	https://teaa.mx/portafolio/ (access on 30 May 2022)
UNITEKBIO PRODUCTS INC.	Blue Cure	Copper sulfate, potassium bicarbonate, potassium silicate, aqueous	https://kr03828467.en.ec21.com/ (access on 30 May 2022)
Vanderbilt Minerals, LLC	Vansil W-10	Calcium metasilicate	https://www.vanderbiltminerals.com/ (access on 30 May 2022)
Yara	ActiSil	Choline chloride and orthosilicic acid	https://www.yara.co.uk/ (access on 30 May 2022)
Organic AgroNutritionals LLC	MegaSilica	Amorphous diatomaceous earth	https://megasilica.com/product/megasilica-organic-soil-conditioner%E2%80%8B-5lbs/ (access on 30 May 2022)

Note: We do not recommend or endorse any of these listed products, product description not available.

**Table 3 biomolecules-12-01027-t003:** Nano silica (Silicon dioxide nanoparticles), nSiO_2_ application for growth and yield attributes in different crops.

Nanoparticle	Concentration (Dose) Applied	Application	Crop	Effect on Plant	Reference
nSiO_2_	60 mg L^−1^	Foliar	Cucumber (*Cucumis sativus*)	Increase in plant height, number of leaves, fruits per plant, fruit length, weight, and yield per plant	[127]
nSiO_2_	600 mg L^−1^, 80 mg Kg^−1^	Foliar, root	Wheat (*Triticum aestivum*)	Enhances the nutrient absorption of N, P, and K and increases the test weight under salinity stress conditions	[128]
nSiO_2_	900 mg L^−1^	Foliar	French bean (*Phaseolus vulgaris*)	Increase in shoot length, number of leaves, leaf area, number of pods, green pod yield, chlorophyll, and carotenoid content	[129]
nSiO_2_	15 kg ha^−1^	Soil	Maize (*Zea mays*)	Enlarges leaf area thus promoting more photosynthetic activity; growth characteristics increase up to 20 days after sowing	[130]
nSiO_2_	2 mM	Foliar	Broad bean (*Vicia faba*)	Increase in pod weight, seed dry weight, and pods per plant under salinity stress conditions	[131]
nSiO_2_	100 ppm	Foliar	Barley (*Hordeum vulgare*)	Improves protein content; enhances yield parameters, such as the number of tillers, plant height, and spike numbers, under water stress conditions	[132]
nSiO_2_	1.5 mM	Foliar	Coriander (*Coriandrum sativum*)	Increase in growth attributes decreased by lead stress; adjusts antioxidant enzyme activities	[133]
nSiO_2_	2 mM	Foliar	Tomatoes (*Solanum lycopersicon*)	Enhances fresh and dry weight of root; increase in root volume and chlorophyll along with photosynthetic rate under salinity stress	[134]
nSiO_2_	3 mM	Soil	Pea (*Pisum sativum*)	Increase in seed weight per pod and total seed yield; enhances potassium ions in roots and shoots; decreases sodium ion concentration; increase in the relative water content	[135]
nSiO_2_, Nano chelated potato specific fertilizer (NPS), Nano chelated complete micron (NCM)	200 gm/100 L	Foliar	Potato (*Solanum tuberosum*)	Increase in tuber yield, dry matter of tubers, starch content, and water use efficiency	[136]
nSiO_2_	1 mM	Foliar	Lentil (*Lens culinaris*)	Increase in seed germination; alleviation of the effect of salinity stress; maintains germination percentage	[137]

## Data Availability

Not applicable.

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
