# Peer review of "Silicon as a Smart Fertilizer for Sustainability and Crop Improvement"

_biomolecules, 2022, doi:10.3390/biom12081027_

Round 1

Reviewer 1 Report

Silicon is an important element for plants. Si supplementation in plants has been extensively studied over the last two decades, and the role of Si in alleviating biotic and abiotic stress has been well documented. This manuscript provide an overview of different smart fertilizer types, application of Si fertilizers in agriculture, availability of Si fertilizers, and experiments conducted in greenhouses, growth chambers, and open fields and also discuss the prospects of promoting Si as a smart fertilizer among farmers and the research community for sustainable agriculture and yield improvement.

In general, this review is well-written and summarize the latest progress in term of Si fertilizer application in crop production, This review can provide an new direction for further research into Si fertilizers as well as promote Si as a smart fertilizer for sustainability and crop improvement, which will provide an important guidance to promote Si fertilizer application in crop production.

Author Response

We would like to thank the worthy reviewer for the time he gives to this manuscript. We highly appreciate your efforts and are very thankful for your positive comments.

Reviewer 2 Report

The manuscript of Tayada et al. is an interesting review about silicon fertilizers and their importance for crop yield improvement. The novelty of this review is focuses specifically on these Si fertilizers
This review contains interesting updated information about Si fertilizers and their application for crop growth and in order to enhance tolerance to abiotic and biotic stresses.
The information is, in general, well organized, with informative figures and tables. But several corrections are required:

1-INTRODUCTION

Sentence from lines 59-61 (Vegetable and fruit crop....in their plant tissues) must be moved to line 38 (after ...to assess the impact on plants.)

3-EFFECTS OF SI FERTILIZERS ON CROPS UNDER CONTROLLED ENVIRONMENTAL CONDITIONS

In this section, some information must be re-ordered. Please put together all information about effect of fertilizers in growth, then on tolerance to abiotic stresses, then on resistance to biotic stresses.

Sentences from lines 132-134 must be put together with sentences from line 147- (just before Monocots can accumulate...)

Lines 162-3: ...including abiotic, biotic STRESS TOLERANCE, and yield enhancement

In Table 2, wheat, effect on plant, delete the final word increases.

4. VARIOUS Si SOURCES FOR Si FERTILIZERS. APPLICATION UNDER FIELD CONDITIONS

The information in this section is a little bit confusing. When explaining about Si sources, it seems that some experiments are under field conditions, and others are in greenhouse/growth chamber conditions. Please, re-order for clarity. Maybe it would be more clear if authors do a section 4. Application under field conditions (containing text from lines 227 to 245) and a
section 5. Various Si sources fro Si fertilizers (containing text from line 187-226).

In Figure 2, some text need correction. Text from (c) (Diatomite....it IS easily soluble...);
(d) (Wollanstonite.....is A naturally occurring mineral also considerED a source...).

In Table 2, it lacks product description form PILARES OPERATION company.

In Table 3, it is not clear for me what is the difference between nSiO2 and Si nanoparticules. Is it the same thing, or different? Please, explain.

In REFERENCES, please check reference style. Most references have the full journal name, others have abbreviated name. Some journal names are with first letters in upper case, and others in lower case. Refs 43, 52, 98, 99, 117, 118, 124-7 lack journal name. Please correct.

OTHER POINTS

In section 2, authors explain about 3 types of fertilizers (nanofertilizer, composite fertilizers,
bioformulation). Please, in section 4.1, explain if explained Si fertilizers are or not composite
fertilizers. It is clear for nanofertilizers (section 4.2)

Author Response

Comments and Suggestions for Authors

The manuscript of Tayada et al. is an interesting review about silicon fertilizers and their importance for crop yield improvement. The novelty of this review is focuses specifically on these Si fertilizers
This review contains interesting updated information about Si fertilizers and their application for crop growth and in order to enhance tolerance to abiotic and biotic stresses. The information is, in general, well organized, with informative figures and tables. But several corrections are required:

Answer: We would like to thank the worthy reviewer for the time he give to this manuscript. All the comments and suggestions were genuine, valuable, and improved the quality of the manuscript. We highly appreciate your efforts and agree to the suggested changes. All the changes that were made in the revised MS can be found with a green color highlight.

1-INTRODUCTION
Sentence from lines 59-61 (“Vegetable and fruit crop....in their plant tissues”) must be moved to line 38 (after “...to assess the impact on plants”.)

Answer: Thank you for the suggestion we have revised as per the suggestion.

3-EFFECTS OF SI FERTILIZERS ON CROPS UNDER CONTROLLED ENVIRONMENTAL CONDITIONS

In this section, some information must be re-ordered. Please put together all information about effect of fertilizers in growth, then on tolerance to abiotic stresses, then on resistance to biotic stresses. Sentences from lines 132-134 must be put together with sentences from line 147- (just before “Monocots can accumulate...”) Lines 162-3: ...including abiotic, biotic STRESS TOLERANCE, and yield enhancement”

Answer: Thank you for the suggestion we have revised as per the suggestion.

In Table 2, wheat, effect on plant, delete the final word “increases”.

Answer: Thank you for the suggestion we have revised as per the suggestion.

  1. VARIOUS Si SOURCES FOR Si FERTILIZERS. APPLICATION UNDER FIELD CONDITIONS
    The information in this section is a little bit confusing. When explaining about Si sources, it seems that some experiments are under field conditions, and others are in greenhouse/growth chamber conditions. Please, re-order for clarity. Maybe it would be more clear if authors do a section 4. Application under field conditions (containing text from lines 227 to 245) and a
    section 5. Various Si sources fro Si fertilizers (containing text from line 187-226).

Answer: Thank you for pointing out this. We have modified and re-arranged the relevant section and hope it more clear now.

In Figure 2, some text need correction. Text from (c) (“Diatomite....it IS easily soluble...”);
(d) (“Wollanstonite.....is A naturally occurring mineral also considerED a source...”).

Answer: Thank you for the suggestion we have corrected the information as per the suggestion.

In Table 2, it lacks product description form PILARES OPERATION company. In Table 3, it is not clear for me what is the difference between “nSiO2” and “Si nanoparticules”. Is it the same thing, or different? Please, explain.

Answer: Sorry for the inconvenience, it’s the same thing, we have addressed the issue and revised it accordingly.

In REFERENCES, please check reference style. Most references have the full journal name, others have abbreviated name. Some journal names are with first letters in upper case, and others in lower case. Refs 43, 52, 98, 99, 117, 118, 124-7 lack journal name. Please correct.

Answer: Thank you for the suggestion we have revised as per the suggestion.

OTHER POINTS
In section 2, authors explain about 3 types of fertilizers (nanofertilizer, composite fertilizers,
bioformulation). Please, in section 4.1, explain if explained Si fertilizers are or not composite
fertilizers. It is clear for nanofertilizers (section 4.2)

Answer: Thank you for the suggestion but we have well-defined and provided overview information in section 2 about smart fertilizers. In 4.1 further, we have not distinguished them we have just provided different sources of Si fertilizers (broadly commercial and noncommercial) in this section.

We have made all the required changes to our manuscript, thanks to the reviewers and editorial team, for their valuable input and time devoted to the manuscript improvement. We hope that with this revision now this manuscript is in a good shape to be published in Biomolecules Journal. 

Kind Regards